# Adaptive Contextual Perception: How to Generalize to New Backgrounds and Ambiguous Objects

**Zhuofan Ying**[1,2]     **Peter Hase**[1]     **Mohit Bansal**[1]
[1]UNC Chapel Hill     [2]Columbia University
{zfying, peter, mbansal}@cs.unc.edu

## Abstract

Biological vision systems make adaptive use of context to recognize objects in new settings with novel contexts as well as occluded or blurry objects in familiar settings [3, 35]. In this paper, we investigate how vision models adaptively use context for out-of-distribution (OOD) generalization and leverage our analysis results to improve model OOD generalization. First, we formulate two distinct OOD settings where the contexts are either *irrelevant* (BACKGROUND-INVARIANCE) or *beneficial* (OBJECT-DISAMBIGUATION), reflecting the diverse contextual challenges faced in biological vision. We then analyze model performance in these two different OOD settings and demonstrate that models that excel in one setting tend to struggle in the other. Notably, prior works on learning causal features improve on one setting but hurt in the other. This underscores the importance of generalizing across both OOD settings, as this ability is crucial for both human cognition and robust AI systems. Next, to better understand the model properties contributing to OOD generalization, we use representational geometry analysis and our own probing methods to examine a population of models, and we discover that those with more factorized representations and appropriate feature weighting are more successful in handling BACKGROUND-INVARIANCE and OBJECT-DISAMBIGUATION tests. We further validate these findings through causal intervention, manipulating representation factorization and feature weighting to demonstrate their causal effect on performance. These results show that interpretability-based model metrics can predict OOD generalization and are causally connected to model generalization. Motivated by our analysis results, we propose new data augmentation methods aimed at enhancing model generalization. The proposed methods outperform strong baselines, yielding improvements in both in-distribution and OOD tests. We conclude that, in order to replicate the generalization abilities of biological vision, computer vision models must have factorized object vs. background representations and appropriately weight both kinds of features.[1]

## 1   Introduction

Humans can recognize objects even when the objects appear in novel settings like new viewpoints, lighting, and background conditions [6, 17]. This kind of generalization has been the subject of many studies in computer vision, where past methods have optimized models to avoid relying on "spurious correlations" (such as the object's background) in order to encourage generalization [46, 2, 38]. However, human vision can also generalize to settings where objects are occluded, blurred, or otherwise distorted, as illustrated by the blurred keyboard in Fig. 1. In fact, research in psychology shows that humans accomplish this task largely *by relying on background information*, which helps disambiguate the identity of the foreground object [24, 53, 39, 23].

---

[1]Our code is available at: https://github.com/zfying/AdaptiveContext

37th Conference on Neural Information Processing Systems (NeurIPS 2023).

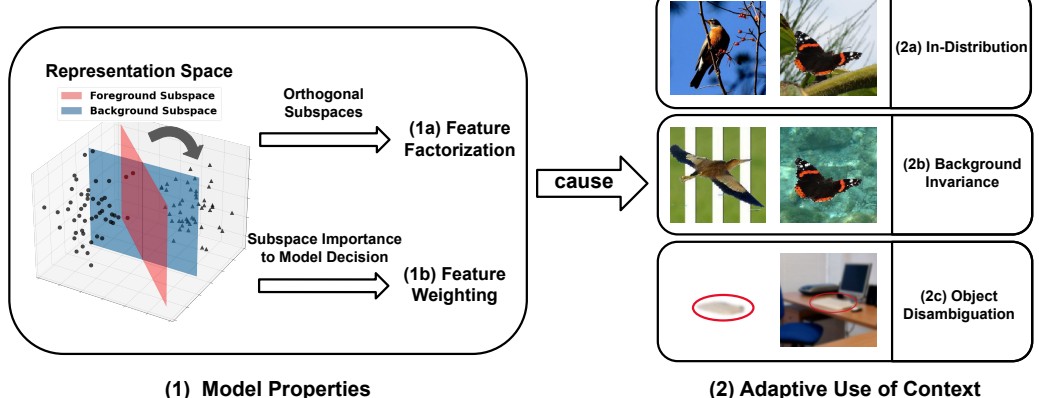

**(1) Model Properties**          **(2) Adaptive Use of Context**

Figure 1: Feature factorization and appropriate feature weighting support Adaptive Contextual Perception. Humans can flexibly generalize to object settings where the context is irrelevant (2b) or helpful (2c). To achieve similar generalization, models should have factorized representations (1a) with strong weights on object features and small but non-zero weights on background features (1b). (2a) In-distribution: In natural environments, foreground and background information correlates strongly. (2b) OOD: BACKGROUND-INVARIANCE. When the background is inconsistent with the foreground object, humans can ignore the background [46]. (2c) OOD: OBJECT-DISAMBIGUATION. When the object is hard to recognize on its own (like the highly blurred keyboard), humans can rely on the background (office setting) to infer what the object is [31].

How can humans both ignore backgrounds to recognize objects in *new contexts* as well as rely on background information to recognize *ambiguous objects*? Past work suggests that this *adaptive use of context* is possible via the *factorization* of foreground objects versus background contexts in visual representations [4, 53, 7, 28]. Factorized representations have the property that the features of the data are represented in different subspaces in the feature space. Specifically, for a model with representations in $\mathbb{R}^d$, with two orthogonal subspaces $V \subset \mathbb{R}^d$ and $V^\perp$, we say the model factorizes its foreground and background representations when foreground information is represented in the subspace $V$ and background information is represented in $V^\perp$.

The bases for the subspaces $V$ and $V^\perp$ do not have to align with the dimensions of the feature vectors (i.e. the standard basis of $\mathbb{R}^d$), nor do representations have to be sparse, unlike common definitions of modularity or disentanglement [8]. Bengio et al. [5] suggest that factorization should broadly improve model generalization; we state this hypothesis specifically for the adaptive use of context as the *Factorization Hypothesis*:

> **Factorization Hypothesis**: factorized representation is necessary for adaptive use of context.

This is not to say that factorized representation is sufficient for generalization. A model must also use the factorized representations in a certain way to generalize to both new contexts and ambiguous objects. In human vision, object-context interactions occur at very small timescales, suggesting simple feedforward weighting of evidence is sufficient [53]. For computer vision, we argue that the object representation typically provides strong evidence for the object's identity, while the context representation can serve as a *tie-breaking* feature that provides a small amount of evidence that is useful when the features of the object itself are ambiguous (all within a single forward pass). In this view, background information becomes useful for identifying objects when they are occluded, blurry, far away, or otherwise difficult to recognize. We state this view as the *Feature Weighting Hypothesis*:

> **Feature Weighting Hypothesis**: model predictions should depend on both object and context features, with stronger weight given to object features but non-zero weight on context features.

In this paper, we argue that models generalize to new contexts and ambiguous objects by following the Factorization Hypothesis and the Feature Weighting Hypothesis. Experimentally, we (1) design testing procedures that capture generalization to both new contexts and ambiguous objects, (2) show that current models' generalization is predicted by how factorized their representations are and how they weight object/context features, and (3) design novel objectives for improving model generalization based on these hypotheses.

First, we formulate two kinds of OOD generalization settings where the context is either irrelevant or helpful: BACKGROUND-INVARIANCE and OBJECT-DISAMBIGUATION, which are illustrated in Fig. 1. In BACKGROUND-INVARIANCE, foreground objects and the background contexts are randomly paired (i.e., objects in new contexts), breaking spurious correlations observed in naturalistic training distributions. In OBJECT-DISAMBIGUATION, the object itself is hard to recognize or ambiguous (due to motion, lighting conditions, noise, etc.), while the background context is of the kind usually observed for the object. We find that there is a fundamental tradeoff in performance on our BACKGROUND-INVARIANCE vs. OBJECT-DISAMBIGUATION tests. This poses a problem for past work, which so far has only evaluated on one setting or the other. We show that methods focusing on BACKGROUND-INVARIANCE [2, 38, 21, 42] will hurt performance on OBJECT-DISAMBIGUATION without realizing it, and vice versa [29]. Following how human vision generalizes, we should aim for a single computer vision system that generalizes well to both OOD settings.

Second, we test the Factorization and Feature Weighting hypotheses by training a population of models on two standard benchmarks, COLOROBJECT and SCENEOBJECT [51, 27]. For each model, we measure factorization using interpretability methods including linear probes, representational similarity analysis [20], and a geometric analysis [28]. To test for feature weighting, we measure model sensitivity to foreground vs. the background. We do so by randomly switching the foreground or background and measuring the relative differences in the changes to model representations. Results show that more factorized models with appropriate feature weighting tend to achieve higher performances on both OOD settings, meaning they adaptively use the context to generalize to novel settings. We further confirm this finding by a causal analysis where we directly manipulate model representations.

Third, we design novel objectives for improving model generalization on BACKGROUND-INVARIANCE and OBJECT-DISAMBIGUATION tests, motivated by the results of our factorization and feature weighting analysis. In order to encourage model factorization, we augment the data with random-background and background-only images (using ground-truth masks or pseudo-masks from SAM [19]), and we weigh the corresponding objective terms to encourage appropriate feature weighting of the foreground and background information. On two benchmark datasets, our method improves significantly over strong baselines on one OOD setting while matching performance on the other.

We summarize our main findings as follows:

1. We show that models that do better on BACKGROUND-INVARIANCE tend to do worse on OBJECT-DISAMBIGUATION and vice versa, highlighting a need to design systems that generalize across both OOD settings, similar to biological vision (Sec. 5).

2. We obtain both correlational and causal evidence for our Factorization and Feature Weighting hypotheses: more factorized models achieve better OOD generalization, and this requires models to assign appropriate weight to foreground features over background features (Sec. 6).

3. We design new model objectives motivated by the Factorization and Feature Weighting hypotheses. We find these objectives achieve a Pareto improvement over strong baselines, increasing performance on one OOD setting while matching performance on the other (Sec. 7).

## 2 Related Works

**Evaluating vision model generalization**. Much research has evaluated the generalization capabilities of vision models in various OOD settings [37, 13, 12]. In particular, several benchmarks have been proposed for evaluating OOD generalization, such as ImageNet-C and ImageNet-9 [15, 46]. However, most prior works fail to test on OOD settings where spurious features like background are beneficial [38, 2, 21]. Liu et al. [29] test on an extreme version of OBJECT-DISAMBIGUATION, where the foreground object is completely masked out, but they do not test on the OOD setting where backgrounds are irrelevant. We evaluate vision models on two kinds of OOD settings where the context is either helpful or irrelevant. To our knowledge, no prior works have evaluated models in both settings.

**Understanding how vision models generalize**. Various works have explored the relationship between model properties, inductive biases, and generalization capabilities [32, 50, 16]. Many recent works try to predict OOD performance using in-distribution (ID) measures like model invariance and smoothness [45, 10, 33], but they do not control for in-distribution accuracy which has historically been the best predictor of OOD accuracy [43]. More recent work that controls for ID accuracy suggests that properties like model invariance, calibration, and robustness do *not* predict OOD generalization

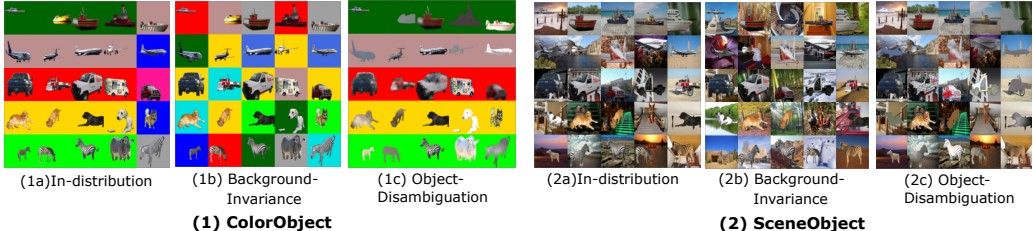

Figure 2: The visualization of COLOROBJECT (left) and SCENEOBJECT (right) datasets and their corresponding OOD test sets. For in-distribution sets (1a, 2a), there are certain natural co-occurrence probabilities between the foreground object classes and the background classes (e.g. 80%, shown by the random background of the last column of (1a, 2a)). For BACKGROUND-INVARIANCE sets (1b, 2b), there is no correlation between the foreground and background. For OBJECT-DISAMBIGUATION sets (1c, 2c), there is a perfect correlation between the foreground and the background classes, but the foreground objects themselves are hard to recognize due to natural corruptions

better than ID accuracy [47]. Moreover, past analysis is generally correlational in nature. In this paper, we use interpretability methods to measure factorization and feature weighting metrics in vision models and analyze both *correlational* and *causal* relations to generalization in our two OOD settings. We do not know of an analysis that demonstrates a causal relationship between model properties and OOD generalization. In their review, Bengio et al. [5] hypothesize that models with more factorized representations should generalize better, but they do not test this experimentally. Our work provides empirical support for this claim specifically for the adaptive use of context for generalization.

**Improving generalization**. Various methods have been proposed to improve the OOD generalization of deep learning models. One line of research relying on metadata of multiple training domains to learn the causal (or 'invariant') features from multiple domains [36, 2, 40, 42]. Other methods make use of data augmentation to improve both ID and OOD performances, like MixUp [52], CutMix [48], and AutoAugment [9]. However, these works do not test on OOD settings where spurious features like background are beneficial. We show in Sec. 5 that methods designed to learn causal features either do not improve or even hurt the performances in the OBJECT-DISAMBIGUATION setting. Our augmentation method is built on the insight that both causal and spurious features can be useful, and these features need to be weighted properly so that the models can generalize to both OOD settings.

## 3 Problem Setup: Two OOD Generalization Settings for Vision Models

Here we introduce the training and testing environments that we would like vision models to perform well in, inspired by biological vision systems. Specifically, we consider scenarios where there are test-time shifts in (1) the correlations between foreground objects and background contexts and (2) the reliability of causal features (features of foreground objects), which can be corrupted by shifts in image lighting, blurring, object occlusion, or other sources of noise. We show an illustrative example in Figure 1: the butterfly is easily recognizable regardless of the background; the keyboard looks like a white blob in isolation but can be easily recognized when put in context. We use train and test sets constructed based on these scenarios to understand how vision systems can effectively use both the causal features of the foreground objects and the spurious features of the background contexts to adaptively generalize to domain shifts.

**Notation.** We use $X$ and $Y$ to denote random variables of the input and the output, while $\mathbf{x}$ and $y$ denote data samples. We assume there is a set of environments $\mathcal{E}$ (instantiated as datasets), which can be divided into in-distribution (ID) and out-of-distribution (OOD) sets: $\mathcal{E}_{id}, \mathcal{E}_{ood} \subset \mathcal{E}$. The models are trained on $\mathcal{E}_{id}$, and our goal is to learn a function $f : X \rightarrow Y$ that generalizes well to $\mathcal{E}_{ood}$.

**Data construction process.** We consider settings where the inputs $X^e$ from environment $e$ are generated from latent variables $Z_c^e, Z_s^e$, where $Z_c^e$ denotes the causal foreground class while $Z_s^e$ denotes the spurious background class. The label $Y^e$ is equivalent to $Z_c^e$ (analogous to e.g. an object recognition task where the label is the object identity). The input $\mathbf{x}$ contains both causal features (foreground object) $\mathbf{x}_c$ and spurious features (background context) $\mathbf{x}_s$. For each sample

$Z_c^e = z_c^e$ and $Z_s^e = z_s^e$, we generate causal and spurious inputs by $\mathbf{x}_c = G_c(z_c^e)$ and $\mathbf{x}_s = G_s(z_s^e)$, where $G_c$ and $G_s$ are maps from the latent space to the input space. We then corrupt the causal features with the random function $\phi$ (for example, by blurring or adding noise to the image). Finally, we generate the complete input $\mathbf{x} = G(\phi(\mathbf{x}_c), \mathbf{x}_s)$, where $G$ combines the causal and spurious features (e.g. overlays a foreground object onto a background image). Each environment differs in its co-occurrence probability $p_{co}$ between $Z_c^e$ and $Z_s^e$, and the level of corruption applied to the causal features $\mathbf{x}_c$. For ID environments $e_{id} \in \mathcal{E}$, we assume high co-occurrence probability and no corruption in the causal feature, identical to past works [38, 51].

**Two OOD Settings.** For OOD environments $e_{ood} \in \mathcal{E}$, we use two particular settings where the spurious background features are either irrelevant or helpful. We call them BACKGROUND-INVARIANCE and OBJECT-DISAMBIGUATION settings, respectively. To create these OOD settings for vision datasets, we need additional annotation of the background classes and the segmentation mask of the foreground objects. We assume we have access to these metadata in all our experiments.

**BACKGROUND-INVARIANCE.** In our BACKGROUND-INVARIANCE setting, foreground objects are randomly paired with background contexts (the co-occurrence probability $p_{co} = 0$). We assume there is no noise to the causal features in this setting ($\phi$ is an identity function). Here, models need to mainly rely on the causal features $\mathbf{x}_c$ to achieve good performances. See Figure 1 and 2 for examples.

**OBJECT-DISAMBIGUATION.** In our OBJECT-DISAMBIGUATION setting, the noise level of the causal features is very high while the co-occurrence probability is also high (like it is in natural images). Here, models need to learn the correspondence between the foreground and background features and make use of the spurious contextual features $\mathbf{x}_s$ to achieve good performance. That is, models must mimic how humans rely on context for object recognition given that objects in the real world can vary significantly in appearance due to factors such as lighting conditions, occlusions, and viewpoint changes [6, 17, 44, 35]. Contextual information, including scene layout, spatial relationships, and semantic associations allows humans to make educated guesses and disambiguate the identity of objects. For example, in Figure 1, the keyboard itself is highly blurred and hard to recognize, but it easily recognizable in context. Also see Figure 2 for examples in our datasets.

# 4 Experiment Setup

**Datasets.** We conduct our analysis and experiments on two standard benchmarks, COLOROBJECT and SCENEOBJECT [38, 51]. The datasets and their experimental settings follow the conventions in the IRM literature [2, 21, 38], but we add a novel OBJECT-DISAMBIGUATION test by using the available metadata. Please refer to Appendix A for further details of the experimental settings.

To set up the COLOROBJECT dataset, following prior works [51], we build a biased dataset using 10 classes of objects selected from MSCOCO dataset [26], which are then superimposed onto 10 colored backgrounds. We set a one-to-one relationship between the foreground classes and the background colors. The co-occurrence probabilities $p_{co}$ are 1.0 and 0.7 for training (i.e., data from these two environments make up the training data), 0.0 for BACKGROUND-INVARIANCE, and 1.0 for OBJECT-DISAMBIGUATION (see Sec. 3 for definitions). To create the OBJECT-DISAMBIGUATION setting for COLOROBJECT dataset where the object itself is hard to recognize, we apply 15 kinds of natural corruptions (like motion blur, contrast change, and fog and snow effects) from Hendrycks and Dietterich [15] to the foreground object before superimposing it on the background image. The natural corruptions come in 5 levels, and we use the average accuracy across 5 levels as the final result. There are 16000 images for training, 2000 for validation, and 2000 for each of the test sets. See Appendix for accuracies across different levels of distribution shift. C. See Figure 2 for examples of ID, BACKGROUND-INVARIANCE, and OBJECT-DISAMBIGUATION images.

To set up the SCENEOBJECT dataset, following prior works [51], we use objects from 10 classes of the MSCOCO dataset and superimpose them onto scene background from 10 classes of the Places dataset to create the SCENEOBJECT dataset [54, 26]. We bind each object category to one scene category. The rest of the data construction is the same as for COLOROBJECT, except for that the co-occurrence probabilities $p_{co}$ are 0.9 and 0.7 for training COLOROBJECT. See Figure 2 for examples.

**Model Population.** Following prior works, we use Wide ResNet 28-2 for the two datasets [49]. Using this model backbone, we train models based on the following methods: empirical risk minimiza-

tion (ERM), IRM[2], VREx[21], GroupDRO[38], Fish[42], MLDG[25], and our own augmentation method (described in Sec. 7). The models are trained on a fixed set of training environments using different loss functions, random seeds, and other hyperparameters. Note that, across datasets, the noise level and the $p_{co}$ in the training sets do not cover the noise level and the $p_{co}$ in the two OOD test sets, so the OOD test sets remain out-of-distribution. We trained a total of 340 models, 170 for each of COLOROBJECT, and SCENEOBJECT datasets. See Appendix A for more details.

## 5 Negative Correlation Between BACKGROUND-INVARIANCE and OBJECT-DISAMBIGUATION Accuracy

**Design.** In this section, we evaluate a population of models in our two OOD settings where the background is either helpful or irrelevant, namely BACKGROUND-INVARIANCE and OBJECT-DISAMBIGUATION. As far as we know, no previous work has evaluated models along both of these dimensions of generalization. We use the population of 170 models for each dataset, described in Sec. 4. We aim to compare how models tend to perform on the two OOD settings.

**Results.** As shown in Figure 3, models that perform better on the BACKGROUND-INVARIANCE setting tend to perform worse on the OBJECT-DISAMBIGUATION setting, and vice versa. The Pearson correlation coefficients are -0.938 for COLOROBJECT and -0.877 for SCENEOBJECT, showing a strong negative correlation between the performances in the two OOD settings. The results are consistent when using a Transformer-based model (Swin) and across training sizes as well (see Appendix Table 7) [30]. Additionally, we find that methods designed to learn causal features, such as IRM, VREx, GroupDRO, Fish, and MLDG, demonstrate improved performance in the BACKGROUND-INVARIANCE setting, but

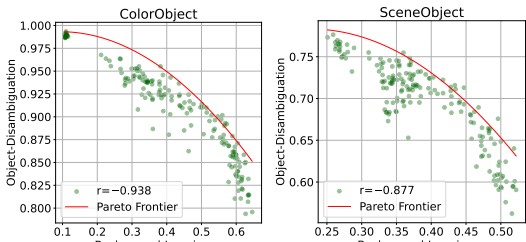

Figure 3: BACKGROUND-INVARIANCE vs. OBJECT-DISAMBIGUATION performances of all models in the COLOROBJECT (left), and SCENEOBJECT (right). Models that do better in one OOD setting tend to do worse in the other, showing strong negative correlations.

they significantly hurt OBJECT-DISAMBIGUATION performance on COLOROBJECT compared to ERM and four of five lower performances on SCENEOBJECT (see Appendix Table 3). These results highlight the need to make a choice about the tradeoff between foreground and background depending on the environment that is expected for deployment.

## 6 Understanding OOD Generalization: Which Vision Models Generalize Better And Why?

In this section, we analyze why some models generalize better than others by testing the Factorization and Feature Weighting hypotheses (see Sec. 1). We do this through a regression analysis that shows that metrics for model factorization and feature weighting help predict OOD generalization, while controlling for in-distribution accuracy and model objectives. We also show that the relationship between the metrics and OOD generalization is causal by directly manipulating feature factorization and feature weighting. Later in Sec. 7, we leverage this analysis to improve model generalization via new objectives. We use the same datasets and population of models as the previous section (Sec. 5).

### 6.1 Regression Analysis

Using our population of models, we a fit simple linear regression that predicts a model's average accuracy between the two OOD settings based on several metrics measured on ID test data. Each metric can be measured for a given hidden layer of a model; we measure each metric across the 3 blocks of Wide-ResNet and include each measurement as a variable in the regression. As a result, we have 9 factorization metrics and 4 feature weighting metrics (see Appendix B for the full list). In total, the regression variables we consider include ID accuracy, model objectives, and our proposed factorization and feature weighting metrics. For each dataset, we run a bootstrap with 10k samples,

using 80% of the 170 models for training and 20% for testing, to obtain an average test $R^2$ regression value that we report along with its standard deviation.

**Factorization Metrics.**    We measure model factorization using three metrics described below. These metrics are computed on representations from a given hidden layer of the model.

1. Linear Probing. To measure factorization with a linear probe, we first create foreground-only and background-only inputs using segmentation masks available in our datasets. We then extract model representations and predict the foreground or background class with a linear model. That is, if there are 10 foreground and 10 background classes, we train a linear classifier to do 20-way classifications. The probe classification accuracy measures factorization because if the model represents foregrounds and backgrounds in overlapping subspaces, the linear classifier will mix up the foreground and background classes, leading to low decoding accuracies [28].

2. Representation Similarity Analysis (RSA). RSA is a method for measuring the separation of inter-class representations and closeness of intra-class representations, which yields more fine-grained information than linear probing [20, 34]. Again, we extract model representations using foreground-only and background-only inputs. Then we compute pairwise similarities of the extracted features with Euclidean distance, giving a *Representation Dissimilarity Matrix (RDM)*. We then calculate the correlation between the model RDM and a predefined *reference RDM*, which represents the ideal factorized representation structure where each foreground/background class has small intra-class distances and large inter-class distances. For details and figure of the reference RDM, see Appendix B.

3. Geometric Analysis. While linear probing and RSA are influenced by class separation *within* each foreground/background subspace, a geometric method may more directly measure factorization independently of this class separation. We adopt the method from Lindsey and Issa [28]. For complete details, see Appendix B, but in short: we compute the factorization of foregrounds against backgrounds as factorization$_{fg} = 1 - \frac{\text{var}_{fg|bg}}{\text{var}_{fg}}$, where var$_{fg}$ denotes the variance in the representation space induced by foreground perturbations and var$_{fg|bg}$ denotes the variance induced by foreground perturbations *within the background subspace*. If representations are factorized, changing foregrounds should not affect the background representations, so var$_{fg|bg}$ should be near zero, and factorization$_{fg}$ will be near 1.

**Feature Weighting Metric.**    We propose a new method, called the Foreground-Background Perturbation Score (FBPS), to measure the relative weight that a model assigns to foreground features vs. background features. First, drawing on ID test data, we create a set of inputs with random foreground and backgrounds. Then, we randomly switch the foreground of each input to get $X_{fg\_flipped}$ and measure the distance in model representations (or outputs) , $\delta_{fg} = \text{dist}(f(X), f(X_{fg\_flipped}))$, with dist as the L2 distance (or KL divergence for model outputs). Similarly, we randomly switch the background contexts to create $X_{bg\_flipped}$ and measure the change in model representations $\delta_{bg} = \text{dist}(f(X), f(X_{bg\_flipped}))$. Finally, we compute the difference in these terms FBPS $= \delta_{fg} - \delta_{bg}$. This approach yields four measurements per ResNet, one per block of the model and one for the model output. This metric is computed and averaged over 5000 samples. See Appendix B for more details.

**Results.**    In Table 1, we report $R^2$ values for regressions predicting the average of the BACKGROUND-INVARIANCE and OBJECT-DISAMBIGUATION accuracies, based on a population of models trained on each dataset. The ID accuracy alone obtains an $R^2$ of 0.191 and 0.691 on the two datasets, respectively. The model objectives serve as a control here, showing significant improvement over ID accuracy alone. Meanwhile, adding either feature factorization or feature weighting metrics can significantly improve $R^2$ in predict-

Table 1: Regression analysis between metrics and the average OOD accuracy. Confidence intervals are one standard deviation over 10k bootstrapping.

|  | $R^2$ predicting average OOD Acc | |
| --- | --- | --- |
| Metric | COLOROBJECT | SCENEOBJECT |
| ID Acc | 0.191±0.150 | 0.691±0.091 |
| + Obj. | 0.505±0.106 | 0.696±0.084 |
| + Obj. & Factor. | 0.925±0.023 | 0.838±0.044 |
| + Obj. & Ft. Wgt. | 0.961±0.013 | 0.857±0.043 |
| + All | **0.963±0.010** | **0.868±0.041** |

ing average OOD accuracies. But **adding feature factorization and feature weighting metrics together achieves extremely high $R^2$ values of 0.963 and 0.868 for predicting OOD accuracy on**

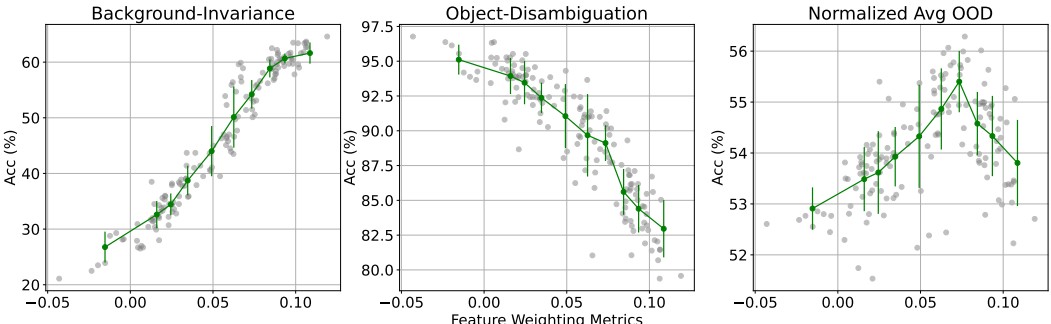

Figure 4: Feature Weighting vs OOD Accuracies. The three plots show the relationship between feature weighting metrics and BACKGROUND-INVARIANCE, OBJECT-DISAMBIGUATION, and the average OOD accuracies in COLOROBJECT. The green lines show the mean and one standard deviation of each bin of data. As models weigh foreground features more, the performance increases in BACKGROUND-INVARIANCE and decreases in OBJECT-DISAMBIGUATION, resulting in an initially increasing and subsequently decreasing average OOD performance. We observe the same trend in SCENEOBJECT as well (see Appendix C).

**the two datasets**. We note that in past studies, various model properties do not predict OOD generalization better than ID acc does [43, 47]. See Appendix C for additional regression visualizations and results in individual OOD settings. Overall, our results suggest that both feature factorization and feature weighting play important roles in models' adaptive use of context.

In Fig. 4, we show the relationship between feature weighting metrics (as measured by FBPS at the final hidden layer of the Wide-ResNet models) and OOD generalization performances in COLOROBJECT. We observe that as the feature weighting metric increases, the performance increases in BACKGROUND-INVARIANCE and decreases in OBJECT-DISAMBIGUATION, resulting in an initially increasing and subsequently decreasing average OOD performance. The trend is the same in SCENEOBJECT too (see Appendix C). This shows that **generalization is maximized at a specific feature weighting level**, suggesting the need to find an appropriate weight between foreground and background features. We show that this relationship is also causal in the next subsection.

## 6.2 Causal Manipulation Analysis

Apart from the regression analysis, we also manipulate factorization and feature weighting in the representational space directly to show their causal effects on performances. We conduct these experiments with 10 ERM models, for both datasets.

**Factorization Manipulation.** Following Lindsey and Issa [28], we directly manipulate the factorization of the model representations. We first identify foreground and background subspaces and then rotate the model representations so that the foreground subspace overlaps more with the background subspace. This transformation directly decreases the feature factorization while holding constant other representation properties that may affect model accuracy [28] (full details in Appendix B). As this intervention applies to the final hidden states of the model, we compute model predictions by applying the linear classifier layer to the transformed representations.

**Feature Weighting Manipulation.** We manipulate the feature weighting by manipulating the variance in a certain subspace, which would bias a linear classifier with L2 regularization to rely on that subspace more (higher variance) or less (lower variance) [14, 41]. We do so by computing the importance, with respect to a subspace of interest (foreground or background), of each feature in the original space and manipulating them by a boost factor $g$. Note this step is followed by retraining the final classifier layer using the adjusted hidden representations. See Appendix B for full details.

**Results.** After we rotate the data to overlap the foreground and background subspaces, feature factorization (measured by the geometric method) drops significantly in SCENEOBJECT, as expected. As a result, the generalization performance in BACKGROUND-INVARIANCE decreases while the

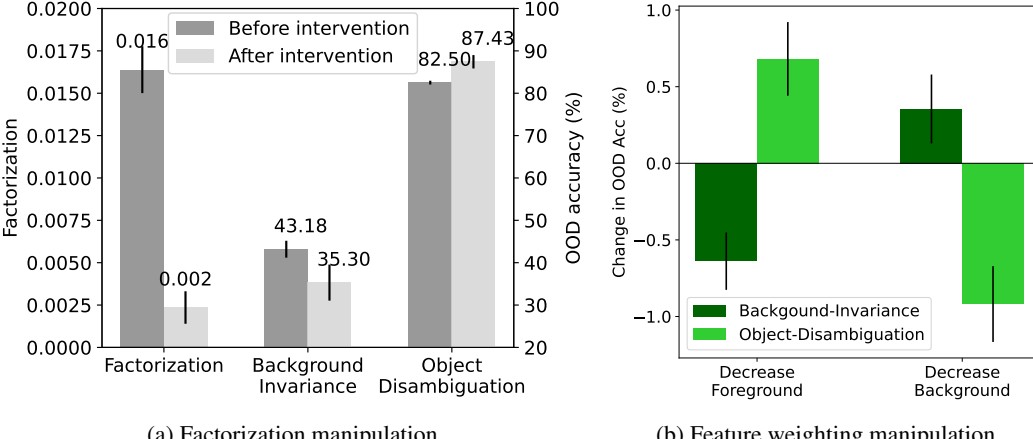

(a) Factorization manipulation  (b) Feature weighting manipulation

Figure 5: Causal intervention results using 10 ERM models in SCENEOBJECT. (a) Rotating the foreground subspace into the background subspace has the effect of reducing model factorization and changing the OOD accuracies in opposite directions. Lighter-colored bars show metrics after the intervention. (b) Decreasing the foreground (or background) subspace by a factor of 64 has the effect of hurting (or improving) BACKGROUND-INVARIANCE accuracy and improving (or hurting) OBJECT-DISAMBIGUATION accuracy. We observe the same trend in COLOROBJECT (see Appendix Fig. 11).

performance in OBJECT-DISAMBIGUATION increases. This is also expected since the less factorized model conflates foreground and background information, and background information is irrelevant for BACKGROUND-INVARIANCE but perfectly correlated with object identity in OBJECT-DISAMBIGUATION. Shown in Fig. 5(a), the factorization metric drop by 85.63% (paired t-test: $p<$1e-8), and the BACKGROUND-INVARIANCE performance drop by 18.24% ($p<$1e-4), while the OBJECT-DISAMBIGUATION performance increases by 5.97% ($p<$1e-5). Therefore, we conclude that **the change in performance is a causal result of decreased factorization**. For results on COLOROBJECT, see Appendix C.

As shown in Fig. 5(b), when we decrease model weight on foreground features, the BACKGROUND-INVARIANCE performance decreases by 0.63 points (paired t-test: $p<$1e-5) while the OBJECT-DISAMBIGUATION performance increases by 0.68 points ($p<$1e-5) in SCENEOBJECT. This shows that increasing the feature weighting of foreground over background *causally* helps in BACKGROUND-INVARIANCE but hurts in OBJECT-DISAMBIGUATION. In the other direction, when we decrease reliance on background features, the BACKGROUND-INVARIANCE performance increases by 0.35 points ($p =$1e-3) while the OBJECT-DISAMBIGUATION performance decreases by 0.92 points ($p<$1e-5), showing that decreasing the feature weighting *causally* hurts BACKGROUND-INVARIANCE generalization but helps in OBJECT-DISAMBIGUATION. These results highlight the dichotomy between the two OOD settings we test, since our interventions improve performance in one setting while hurt in the other. This trend is expected since the background is irrelevant in BACKGROUND-INVARIANCE (where decreasing the background importance helps), while the background is helpful in OBJECT-DISAMBIGUATION (where decreasing background importance hurts). Overall, our results show the causal effect of feature weighting on model generalization, further strengthening our hypothesis.

# 7 Proposed Methods: Objectives for Adaptive Contextual Perception

Finally, in this section, we propose an augmentation method for improving the generalization ability to both OOD settings, based on our analysis in Sec. 6. Like prior methods [18, 11], our approach leverages additional annotations, including background identity and segmentation masks of foreground objects, which are available in our COLOROBJECT and SCENEOBJECT datasets. We also experiment with using Segment-Anything (SAM) to automatically generate segmentation masks to alleviate the need for additional annotation and achieve similar performances (see Appendix C.4 [19].

Table 2: Accuracy of our proposed method (plus/minus one standard deviation over 10 seeds). OOD1 denotes BACKGROUND-INVARIANCE, and OOD2 denotes OBJECT-DISAMBIGUATION

| | COLOROBJECT | | | | | |
| Method | ID | OOD1 | OOD2 | Avg OOD | Factorization | FBPS |
| --- | --- | --- | --- | --- | --- | --- |
| ERM | 87.15±0.40 | 29.11±3.39 | **95.24±0.83** | 62.17±1.40 | 71.27±1.14 | 0.012±0.013 |
| Best Baseline | **88.48±0.85** | **41.13±7.27** | 92.65±2.37 | **66.89±2.73** | 72.87±1.58 | 0.044±0.027 |
| Ours | 87.45±0.48 | **42.97±3.02** | 92.02±1.06 | **67.50±1.80** | **76.93±1.33** | 0.047±0.018 |

| | SCENEOBJECT | | | | | |
| Method | ID | OOD1 | OOD2 | Avg OOD | Factorization | FBPS |
| --- | --- | --- | --- | --- | --- | --- |
| ERM | 72.28±0.93 | 35.37±1.97 | 71.93±1.27 | 53.65±0.70 | 64.07±0.98 | 0.000±0.006 |
| Best Baseline | 73.47±0.79 | **36.91±1.49** | 71.30±0.74 | 54.11±0.61 | 64.95±0.85 | 0.005±0.005 |
| Ours | **74.13±0.48** | 36.83±1.34 | **73.86±1.11** | **55.34±0.51** | **68.22±1.01** | 0.011±0.008 |

**Augmentation Objectives.** Our augmentation method consists of two parts: *random-background* and *background-only* augmentation. First, we extract the foreground object from an image and combine it with a randomly chosen background segmented from another image in the batch. This process encourages the model to recognize the object regardless of the background.

The cross-entropy loss for these images is denoted as $\mathcal{L}_{fg}$. Second, we create background-only images by replacing the foreground objects with black pixels and then ask the model to predict background labels. This prevents the model from ignoring all backgrounds so that it can still rely on the background when the foreground object is hard to recognize. We denote cross-entropy loss for background-only images as $\mathcal{L}_{bg}$. The final loss is: $\mathcal{L} = \alpha_0 \mathcal{L}_{task} + \alpha_1 \mathcal{L}_{fg} + \alpha_2 \mathcal{L}_{bg}$, where $\mathcal{L}_{task}$ is the task loss and $\alpha_0, \alpha_1$ and $\alpha_2$ are scalar weights for different losses. By adjusting the value of $\alpha_0, \alpha_1$, and $\alpha_2$, we can find the optimal balance among the original task loss and two augmentation types, enabling the model to learn more effectively from both background and foreground features and adaptively generalize to the two OOD settings. See Appendix A for method and tuning details.

**Results.** Our proposed augmentation method demonstrates significant improvements in both ID and OOD accuracies compared to ERM and the best baseline, Fish [42]. On COLOROBJECT, we match the best baseline on both OOD settings. Here, ERM still does better on OBJECT-DISAMBIGUATION, but on average OOD accuracy, our method improves over ERM by over 5.6 points, matching the best baselines. On SCENEOBJECT, we match the best baseline on BACKGROUND-INVARIANCE while significantly improving over it by 2.5 points on OBJECT-DISAMBIGUATION. We also improve upon ERM by 1.9 points on OBJECT-DISAMBIGUATION, where no other baseline methods can beat ERM. We also improved the average OOD performance by 1.2 points over ERM and other baselines. We get similar improvements over baselines for our method with pseudo-masks generated by SAM (see Appendix Table 6). And the improvements are even more significant with smaller training sizes (also see Appendix Table 6). In both datasets, our method improves over ERM or the best baseline on feature factorization and feature weighting, suggesting a causal relationship between these model properties and model generalization. For results of varying combination weights for the two augmentations, see Appendix C.4.

## 8 Conclusion

In summary, we find that: (1) Vision models face a tradeoff in performing well on BACKGROUND-INVARIANCE and OBJECT-DISAMBIGUATION tests. By evaluating a population of 170 models, we find a strong negative correlation coefficient of about $r = 0.9$ between the accuracies for each OOD test. (2) OOD generalization in models is explained by the Factorization Hypothesis and the Feature Weighting Hypothesis. In our correlational analysis, metrics for factorization and feature weighting are highly predictive of OOD generalization ($R^2$ up to .96, controlling for in-distribution accuracy). In our causal analysis, we find that directly manipulating factorization and feature weighting in models allows us to control how well they generalize. (3) Lastly, we propose data augmentation methods that achieve Pareto improvements over strong baselines, as they are able to improve either BACKGROUND-INVARIANCE or OBJECT-DISAMBIGUATION without hurting the other setting.

## Acknowledgements

We thank Jack Lindsey and Elias Issa for valuable discussion regarding the causal analysis. This work was supported by ARO Award W911NF2110220, DARPA Machine-Commonsense (MCS) Grant N66001-19-2-4031, ONR Grant N00014-23-1-2356, DARPA ECOLE Program No. HR00112390060, and a Google PhD Fellowship. The views contained in this article are those of the authors and not of the funding agency.

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

# A  Experiment Design Details

## A.1  Dataset Details

**Dataset License.**  Our datasets are created using MSCOCO and Places [26, 54], both under the CC BY 4.0 license.

**Synthetic Dataset.**  We design a synthetic dataset for a clear demonstration of our hypothesis and results.  Following the data generation process formulated in Sec. 3, we create synthetic data with distribution shifts in co-occurrence probability and noise in foreground features.  The ground truth label $Y$ and latent variables $Z_c, Z_s$ are all in $\{0, 1\}$. $Y$ and $Z_c$ is sampled uniformly: $Y = Z_c \sim U(\{0, 1\})$. The causal features $\mathbf{x}_c$ and spurious features $\mathbf{x}_s$ are generated from Gaussian distributions based on $Z_c$ and $Z_s$ respectively: $x_c \sim \mathcal{N}(\mu_c^i, \sigma_c^2)$ for $z_c = i$, and $x_s \sim \mathcal{N}(\mu_s^i, \sigma_s^2)$ for $z_s = i$. We choose continuous features and Gaussian distributions instead of binary features and Bernoulli distributions like prior works [51, 27] to model noise and the OBJECT-DISAMBIGUATION setting more easily.  We set $\mu_{r0} = \mu_{nr0} = 0.2$, $\mu_{r1} = \mu_{nr1} = 0.8$, and $\sigma_r = \sigma_{nr} = 0.2$. The dimensions of inputs are $80 + 10 + 10 = 100$ for useless, robust, and non-robust features respectively. We then add noise to causal features: $\epsilon \sim \mathcal{N}(0, \sigma_\epsilon^2)$. Finally, we have irrelevant features $\mathbf{x}_i$ that do not correlate with the label and are sampled from another Gaussian distribution: $x_n \sim \mathcal{N}(\mu_n^i, \sigma_n^2)$. Finally, we combine the noisy causal features, spurious features, and irrelevant features to create the complete inputs. In training sets, the co-occurrence probability $p_{co}$ varies between $[0.5, 0.98]$ and the noise level $\sigma$ between $[0.6, 1.6]$ (see Sec. 3 for definitions). Each train set has 1000 samples. In BACKGROUND-INVARIANCE setting, $p_{co} = 0$ and $\epsilon = 0$, while in OBJECT-DISAMBIGUATION setting, $p_{co} = 1$ and $\epsilon = 1.5$.

**COLOROBJECT.**  Following Ahmed et al. [1]and Zhang et al. [51], we superimpose objects from 10 classes in MSCOCO dataset onto 10 colors. The RGB values of the predefined colors are: [0, 100, 0], [188, 143, 143], [255, 0, 0], [255, 215, 0], [0, 255, 0], [65, 105, 225], [0, 225, 225], [0, 0, 255], [255, 20, 147], [160, 160, 160]. The object classes are: boat, airplane, truck, dog, zebra, horse, bird, train, bus, motorcycle. Each image has the size $3 \times 64 \times 64$, and the dataset sizes are 8000 for the train set and 1000 for validation, ID test, and the two OOD test sets. Since we use two environments in training ($p_{co} = 0.9, 0.7$), models see 16000 images during training. In the BACKGROUND-INVARIANCE setting, $p_{co}$ is set to 0.0. In the OBJECT-DISAMBIGUATION setting, $pco$ is set to 1.0, and we apply 15 kinds of natural corruption, each with 5 levels of severity, to the foreground images before superimposing them with the backgrounds.

**SCENEOBJECT.**  Following Ahmed et al. [1], Zhang et al. [51], we superimpose objects from 10 classes in MSCOCO dataset onto 10 scene classes in Places dataset. The object classes are the same as COLOROBJECT, and the background scene classes are: beach, canyon, building facade, staircase, desert sand, crevasse, bamboo forest, broadleaf, ball pit, and kasbah. The co-occurrence probabilities $p_{co}$ are set to 1.0 and 0.7. The image size and dataset sizes are the same as COLOROBJECT as well.

## A.2  Model and Training Details

**Model Architecture and Hyperparameters.**  For the synthetic dataset, we use simple MLPs with 3 hidden layers and 5 neurons each hidden layer with ReLU activation. We train each model for 750 epochs with SGD with a learning rate of 0.1. The learning rate is decreased by a factor of 10 every 250 epochs. For COLOROBJECT and SCENEOBJECT, we use Wide ResNet 28 following prior works [49]. To adapt to the 64 by 64 image size, we change the average pooling layer window size from 8 to 16. The learning rate is set to 0.1, which is by a factor of 10 for every 1500 update steps. The model is updated with 4000 steps in total. The batch sizes are 128 and 64 for COLOROBJECT and SCENEOBJECT respectively.

**Tuning for Proposed Augmentation Method.**  We tune the weights for different objectives by fixing $\alpha_0 = 1$ and $\alpha_1 + \alpha_2 = 1$ so that we can keep the same learning rate. We train on all combinations with a stride of 0.1 for $\alpha_1, \alpha_2$ (ranging from 0.0 to 1.0). This results in 11 conditions, and we train 10 models with different random seeds. Model checkpoints are selected based on in-distribution validation set accuracy. To pick one condition to report in Sec. Sec. 7 and Table 2, we pick hyperparameter values for which our method matches the performance of the best baseline on one

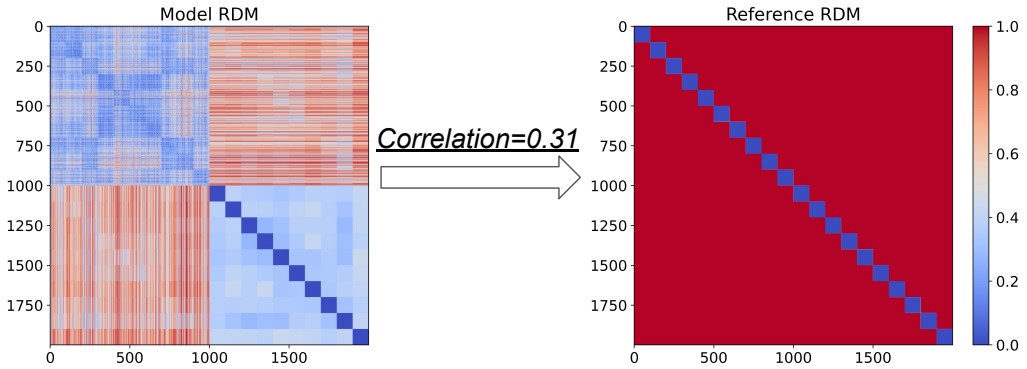

Figure 6: Framework of RSA. We compute the model RDM and its correlation with the reference RDM, which is used as a metric for factorization. Each cell in the RDM represents the dissimilarity between representations of two images (the value at the position $(i, j)$ represents the dissimilarity between representations of image $i$ and image $j$). The diagonal cells represent the dissimilarity of each image with itself, which is always zero.

OOD test while improving on the other (i.e. achieves a Pareto improvement). These hyperparameters are $\alpha_1 = 0.7$ and $\alpha_2 = 0.3$ in both datasets.

**Model Population.** For the synthetic dataset, we train 196 models in total. We vary the co-occurrence probability $p_{co}$ in [0.7, 1.6] and noise level in [0.5, 0.98], yielding 200 environments, and we train one model per environment. For COLOROBJECT and SCENEOBJECT, we train 170 models in total. All methods are trained with 10 random seeds. Therefore, we have 60 models for the 6 baseline methods (ERM, IRM[2], VREx[21], GroupDRO[38], Fish[42], and MLDG[25]). For our proposed augmentation method (see Sec. 7), we fix $\alpha_1 + \alpha_2 = 1$ so that we can keep the learning rate the same. We then vary $\alpha_1$ from 0 to 1 with 0.1 as the intervals and train 10 models for each, which gives us 110 models.

**Training Environments.** All experiments are conducted using Nvidia RTX 2080 Ti. It takes about half an hour to train one Wide ResNet model. Simple MLPs are trained on CPUs.

## B  Analysis Details

### B.1  Regression Analysis Details

We use 9 factorization metrics and 4 feature weighting metrics to predict OOD performances in a linear regression analysis. We measure the three factorization metrics (linear probing, RSA, and geometric methods) across the 3 blocks of Wide-ResNet and our simply MLPs, giving us 9 metrics in total. We also measure the FPPS metrics across the 3 blocks *and* the final output probability layer. We use L2 distance for the 3 blocks and KL divergence for the output layer, giving us 4 metrics in total. These metrics are used to predict OOD performances with 10k booststrapping.

**Factorization Metrics Details.**

1. **Linear Probing**. We first create foreground-only and background-only images using ID test set data. We then extract model representations and predict the foreground or background class with a linear model. That is, if we have $n$ foreground classes and $n$ corresponding background classes in total, we do $2n$-way classification. We name this probing method $probe_{2n}$. The probing dataset size is therefore 2000, twice the size of the ID test set. We compute average accuracy with 5-fold cross-validation and use it as the measurement of feature factorization.

2. **RSA Analysis**. We first extract foreground-only and background-only features using ID test data with additional annotation. Then we compute pairwise similarities of the extracted features with Euclidean distance, giving a model *Representation Dissimilarity Matrix (RDM)* (see Fig. 6 for a visualization). The RDMs have a size of $n$ by $n$, where $n$ denotes the number of images in

the dataset. The value at position $(i, j)$ is the Euclidean distance between image $i$ and image $j$ in the representational space. The diagonal values represent the dissimilarity of each image with itself, which is always zero. The reference RDM shows an ideal representational geometry. In the case of factorization, we want the images of the same class to have similar representations (dissimilarity=0), while images of different classes to have different representations (dissimilarity=1). Similar to the probing method, foreground and background classes are regarded as different classes in order to test the ability of the models to separate foreground features from background features (e.g. 20 classes in total for our COLOROBJECT and SCENEOBJECT datasets). We then compute the Pearson correlation between the model RDM and the reference RDM and use the $r$ number as a metric for factorization. Note that since we do not normalize the model RDM, the absolute value of $r$ is meaningless. In a regression analysis where we are trying to predict OOD generalization, the relative differences among the $r$ values are what matters.

3. **Geometric Analysis**. Following Lindsey and Issa [28], we compute the factorization of foregrounds against backgrounds as factorization$_{fg} = 1 - \frac{\text{var}_{fg|bg}}{\text{var}_{fg}}$, where var$_{fg}$ denotes the variance in the representation space induced by foreground perturbations and var$_{fg|bg}$ denotes the variance induced by foreground perturbations within the background subspace.

To get var$_{fg}$, we compute the variance induced by the foreground perturbations. Specifically, for each of the 1000 test background images, we pair it with 100 random foregrounds and obtain model representations for each combined image. For each background image, we compute the scalar variance induced by the 10 random foregrounds by first computing the variance within each vector dimension and then summing over all dimensions. Then we average this per-background-image variance over the 1000 backgrounds to get var$_{fg}$. Second, to compute var$_{fg|bg}$, we need to find a subspace for background information. To do so, we use the same set of background images paired with 10 random foreground images. We obtain a special set of background representations from 1000 test background images, each paired with 10 random foregrounds. The representations are the average model representation per background image, averaged across the image's 10 random foregrounds. Then, we conduct PCA on this set of representations, and we select the top-$k$ principal components to get a subspace $U_{bg}$ that explains 99% of the variance in the representations. Finally, to get var$_{fg|bg}$, we project the representations of the 100,000 background plus random foreground images (same as used to compute var$_{fg}$) into the background subspace. For these projected representations, we repeat the process of computing a single scalar variance (as with var$_{fg}$) to obtain the variance induced by foreground perturbation in the background subspace, var$_{fg|bg}$.

## B.2 Causal Analysis Details

**Manipulating Factorization** We first compute the top-$k$ PCs of the mean representation of each foreground class so as to explain 99% of the variance, referred to as the inter-class PCs $U_{inter}$. Specifically, for each class, we have 100 foreground images, each randomly paired with 100 backgrounds. The mean representation of each class (class center) is the average representation of the 10000 images. Then we also computed the PCs of the data with corresponding class centers subtracted from each activity pattern, referred to as the intra-class PCs $U_{intra}$. Specifically, for each datapoint $\mathbf{x}$, we subtract its corresponding class center from it: $\mathbf{x}' = \mathbf{x} - c_i$, where $c_i$ is its class center. Then we compute $U_{intra}$ with all subtracted datapoints. Then, we transformed the data by applying to the class centers a change of basis matrix that rotated each inter-class PC into the corresponding (according to the rank of the magnitude of its associated eigenvalue) intra-class PC: $U = (U_{intra}[: num\_of\_classes])^T * U_{inter}$, where $U$ is the desired transformation, the rows of $U_{inter}, U_{intra}$ are sorted according to their eigenvalue, and $num\_of\_classes$ is the number of classes (10 in our cases). We limit the intra-class PCs to its top-$num\_of\_classes$ PCs since the number of classes is much smaller than the dimensionality of the extracted features and $U_{inter}$ has much fewer rows than $U_{intra}$. This transformation has the effect of decreasing factorization while controlling for all other statistics of the representation that may be relevant to object classification performance (like invariance). Finally, after transforming the data by $\mathbf{x} * U^T$, we can measure the OOD performances and the factorization metrics and observe the causal relationship between them.

**Manipulating Feature Weighting.** To manipulate feature weighting, we first compute the subspace of interest (foreground or background) using PCA. Specifically, similar to the geometric method of measuring factorization, we compute the average representations of foreground/background images by averaging it over 100 images paired with random backgrounds/foregrounds. We then conduct

Table 3: OOD evaluation for baseline models. OOD1 refers to BACKGROUND-INVARIANCE, while OOD2 refers to OBJECT-DISAMBIGUATION. Prior methods focusing on learning causal features consistently improve generalization on BACKGROUND-INVARIANCE settings, but are unhelpful or even hurt performances on OBJECT-DISAMBIGUATION settings. Confidence intervals are one standard deviation across 10 seeds.

| | COLOROBJECT | | | SCENEOBJECT | | |
|---|---|---|---|---|---|---|
| | ID Acc | OOD1 Acc | OOD2 Acc | ID Acc | OOD1 Acc | OOD2 Acc |
| ERM | 87.15±0.40 | 29.11±3.39 | **95.24±0.83** | 72.28±0.93 | 35.37±1.97 | 71.93±1.27 |
| IRM | 87.10±0.58 | 31.68±1.70 | 94.15±0.82 | 71.53±0.69 | 34.33±0.97 | 70.98±1.37 |
| VREx | 87.60±0.48 | 36.38±3.16 | 92.65±0.77 | 72.01±0.56 | 35.43±1.15 | 71.27±1.09 |
| GroupDRO | 87.47±0.70 | 40.92±3.72 | 90.58±2.06 | 69.71±0.67 | 34.18±0.96 | 68.19±1.15 |
| Fish | **88.48±0.85** | **41.13±7.27** | 92.65±2.37 | 73.47±0.79 | **36.91±1.49** | 71.30±0.74 |
| MLDG | 87.90±0.39 | 36.54±2.63 | 93.30±1.92 | **73.79±0.63** | 36.59±1.01 | 72.26±0.82 |

PCA on the average representations for both the foregrounds ($U_{fg}$) and the backgrounds ($U_{bg}$). Then, we compute the importance relative to a subspace of interest (foreground or background) $e_i$ for each feature $\mathbf{x}_i$ in the original space, where $i$ is the index of the feature. To do so, we sum up the absolute weights in the PCA transformation matrix weighted by the ratio of variance explained: $e_i = \sum_j U_{fg}[j, i] * \sqrt{r_j}$, where $U_{fg}[j, i]$ refers to the $i$-th weight of the $j$-th PC component and $r_j$ refers to the ratio of variance explained by the $j$-th PC component. We then normalize $e_i$ and multiply them by a factor $g$ to get the boost weights for each feature in the original space. Finally, we get the boosted feature $\mathbf{x}_{boosted} = \mathbf{x} * e * g$.

Since changing the variance of the data already has an effect on the accuracy of a linear classifier because of the L2 regularization, we control for this effect by computing the accuracy of boosting all features uniformly: $\mathbf{x}_{control} = \mathbf{x} * g$. We report the differences in 5-fold cross-validation performances of linear classifiers trained on $\mathbf{x}_{boosted}$ versus $\mathbf{x}_{control}$. Note that if the foreground and background subspaces are overlapping, then boosting one will boost the other as well. But as long as the two subspaces are not completely overlapping, we can boost or decrease one of them more than the other.

## C    Additional Results

### C.1    OOD Evaluation

**Methods focusing on learning causal features improve performance on BACKGROUND-INVARIANCE but lower perfomance on OBJECT-DISAMBIGUATION.** As shown in Table 3, methods designed to learn causal features (e.g. IRM, VREx, GroupDRO, Fish, and MLDG) demonstrate improved performance in the BACKGROUND-INVARIANCE setting, but they significantly hurt OBJECT-DISAMBIGUATION performance on COLOROBJECT compared to ERM and four of five lower performances on SCENEOBJECT.

**OOD Evaluation with various training sizes.** We evaluate performances with various training sizes (4k, 8k, 16k). As shown in Figure 7(a), we see clear Pareto frontiers across various training sizes. In addition, the Pareto frontiers move outwards as the training size gets larger. With 8k data, the frontier is close to the frontier with 16k data, suggesting that the frontier is close to saturation and that our argument is unlikely to be restricted by our training size.

**OOD Evaluation on transformer architecture.** We also evaluate performances on vision transformers (Swin-tiny) on the two OOD settings in SCENEOBJECT [30]. Since applying domain generalization methods (like Fish) to transformers requires a significant amount of hyperparameter tunings, we only evaluate a population of models trained with our augmentation method and baseline ERM. As shown in Figure 7(b), we see clear trade-offs between BACKGROUND-INVARIANCE and OBJECT-DISAMBIGUATION for Swin-tiny, despite that they have significantly better performances than ResNets. This suggests that our argument is unlikely to be restricted to particular architecture choices.

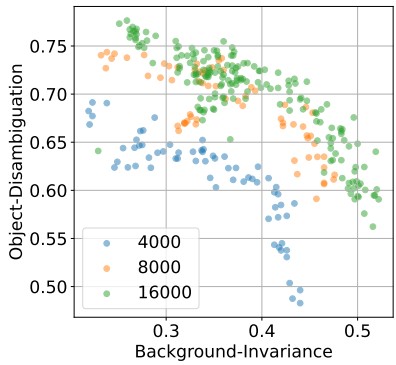
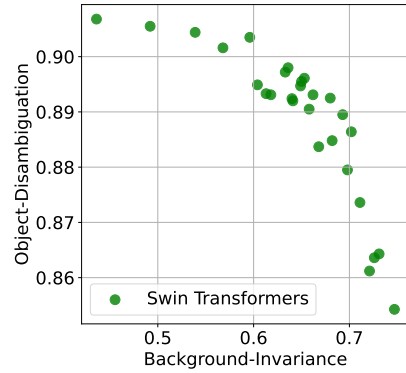

(a) Pareto Frontier with Various Training Sizes    (b) Pareto Frontier with Vision Transformers

Figure 7: Pareto frontier on SCENEOBJECT. We see clear trade-offs between BACKGROUND-INVARIANCE and OBJECT-DISAMBIGUATION across different training sizes (4k, 8k, 16k) and for vision transformers (Swin-tiny) [30].

**Performances Across Different Levels of OOD.** We present the results of baseline models (ERM, IRM, VREx, GroupDRO, Fish, and MLDG) across varying degrees of OOD settings in Fig. 8. For BACKGROUND-INVARIANCE, we run tests on varying levels of $p_{co} = 0.6, 0.45, 0.3, 0.15, 0.0$. Here, $0.0$ is the setting that we report in all other sections. Importantly, all these settings are considered OOD as the training environment $p_{co}$ is always at least $0.7$. For OBJECT-DISAMBIGUATION, we present results on five distinct levels of corruption. The level of corruption here refers to the severity of the deviation from the original distribution, where a higher level signifies a more extreme corruption. We report the mean performance across these five levels in all other sections of this study. Across these diverse OOD settings, we observe that the performance of each model decreases as the settings become increasingly OOD. Consistent across different levels of OOD, other baselines can improve over ERM in BACKGROUND-INVARIANCE, but generally hurts the performance in OBJECT-DISAMBIGUATION, highlighting a need to design systems that generalize across both OOD settings, similar to biological vision.

## C.2 Regression Analysis

**Full Regression Results.** We use 9 factorization metrics and 4 feature weighting metrics in total for our regression analysis. We use all metrics, together with the ID accuracy and model objectives, to predict OOD generalization. To visualize our regression analysis results, as shown in Fig. 9, we combine all metrics into one single metric that correlates strongly with the average OOD performance. This single metric is obtained by combining all metrics with weights from the linear regression that predicts the average OOD accuracy. The Pearson correlation between the average OOD accuracy and the combined metrics has $r$ values of $0.926, 0.986, 0.952$ for the synthetic dataset, COLOROBJECT, and SCENEOBJECT respectively. The regression weight and $p$ value for each individual metric are shown in Table 5. Note that different metrics have different scales, so the absolute values of the coefficients are not comparable to each other. The bold numbers are ones with $p$-values smaller than 0.05. ID accuracy, as well as our feature weighting metrics FBPS, is important in all three datasets for predicting OOD generalization. All three kinds of metrics of feature factorization are useful in some cases (the RSA and geometric methods in COLOROBJECT and the probing method in SCENEOBJECT). As shown in Table 4, our factorization and feature weighting metrics significantly improve the ability to predict performances in all three settings: BACKGROUND-INVARIANCE, OBJECT-DISAMBIGUATION, and average OOD.

**Feature Weighting Results on COLOROBJECT.** We observe the same trend in SCENEOBJECT (in Fig. 10) as in SCENEOBJECT (in Fig. 4). As models weigh foreground features more, the performance increases in BACKGROUND-INVARIANCE and decreases in OBJECT-DISAMBIGUATION, resulting in an initially increasing and subsequently decreasing average OOD performance.

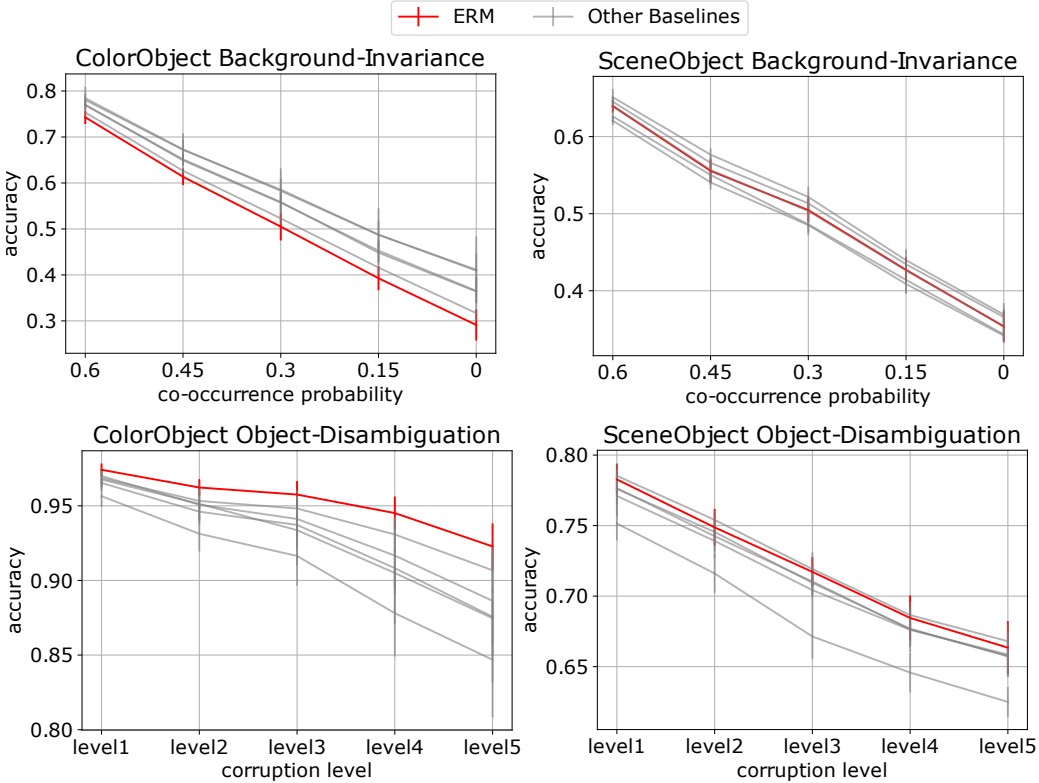

Figure 8: Performances across Different Levels of OOD settings. Confidence intervals are one standard deviation across 10 random seeds. As the test sets become more OOD, the performances for all methods drops (training co-occurence is 0.7, e.g.). Other baselines sometimes improve in BACKGROUND-INVARIANCE, but consistently hurt the performances in OBJECT-DISAMBIGUATION.

Table 4: $R^2$ for the regression analysis between metrics and BACKGROUND-INVARIANCE, OBJECT-DISAMBIGUATION, and the average OOD accuracy. Confidence intervals are one standard deviation over 10k bootstrapping. Bold numbers are $p<0.05$ with independent t-test.

| Metric | COLOROBJECT | | | SCENEOBJECT | | |
|---|---|---|---|---|---|---|
| | OOD1 | OOD2 | Avg OOD | OOD1 | OOD2 | Avg OOD |
| ID Acc | 0.162±0.137 | 0.067±0.118 | 0.191±0.150 | 0.238±0.122 | -0.036±0.078 | 0.691±0.091 |
| + Obj. | 0.505±0.095 | 0.413±0.103 | 0.505±0.106 | 0.271±0.121 | 0.012±0.104 | 0.696±0.084 |
| + Factor. | 0.952±0.014 | 0.900±0.030 | 0.925±0.023 | 0.780±0.060 | 0.798±0.048 | 0.838±0.044 |
| + Ft. Wgt. | 0.981±0.006 | 0.856±0.033 | 0.961±0.013 | 0.977±0.006 | 0.905±0.027 | 0.857±0.043 |
| + All | **0.985±0.004** | **0.915±0.024** | **0.963±0.010** | **0.978±0.006** | **0.936±0.021** | **0.868±0.041** |

## C.3 Causal Analysis

**Causal Analysis Results on COLOROBJECT.** For the causal analysis on factorization in COL-OROBJECT, we see a similar trend to SCENEOBJECT in that both the factorization (as measured by the geometric method) and the BACKGROUND-INVARIANCE performance decrease significantly. However, the OBJECT-DISAMBIGUATION accuracies remain the same as opposed to increasing. This is because the decoding accuracies of linear classifiers on backgrounds are near the ceiling (100%) both before and after the causal intervention, which is a result of the backgrounds (10 pure colors) being too easy to decode. For the causal analysis on feature weighting in COLOROBJECT, we observe the same trend as in SCENEOBJECT: when we decrease the foreground (or background) features, the BACKGROUND-INVARIANCE performances decrease (or increase) while the OBJECT-DISAMBIGUATION performances increase (or decrease). All results are significant (paired t-test $p<1e-3$).

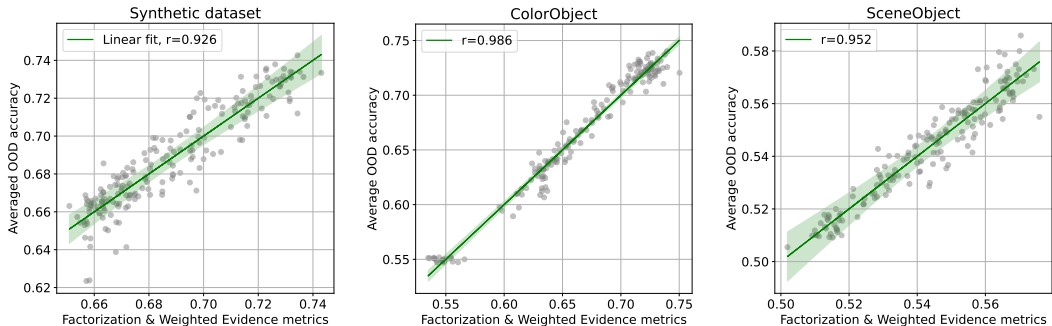

Figure 9: Correlation between combined metrics and combine OOD performance. A dot donates a model. The $x$ axis is the combined metrics, including ID accuracy, objectives, and all factorization and feature weighting metrics. These metrics are combined with weights from a linear regression. The $y$ axis is the average OOD performance from the two settings. The green lines are fit by linear regressions, and the shaded areas are 95% confidence intervals. We can observe strong correlations between the combined metrics and the combined OOD performances across the two datasets.

Table 5: Regression Weight and $p$ Value. We have 9 metrics for measuring factorization and 4 metrics for feature weighting. The $p$-values are put in brackets. Confidence intervals are one standard deviation across 10k bootstrapping. Bold numbers are ones with $p$-value smaller than 0.05. Note that different metrics have different scales, so the absolute values of the coefficients are not directly comparable.

| | Regression Coeffcients & $p$-value | | |
| --- | --- | --- | --- |
| | Synthetic | COLOROBJECT | SCENEOBJECT |
| ID Acc | 0.044±0.039 ($p$=0.242) | **1.803±0.152 ($p$=0.000)** | **0.782±0.059 ($p$=0.000)** |
| probing_layer1 | **-0.057±0.019 ($p$=0.005)** | -0.151±0.108 ($p$=0.160) | -0.027±0.029 ($p$=0.357) |
| probing_layer2 | 0.039±0.030 ($p$=0.179) | -0.200±0.104 ($p$=0.060) | **0.128±0.038 ($p$=0.001)** |
| probing_layer3 | -0.008±0.026 ($p$=0.762) | -0.013±0.117 ($p$=0.907) | **0.338±0.053 ($p$<0.001)** |
| RSA_layer1 | **0.126±0.025 ($p$<0.001)** | **0.172±0.039 ($p$<0.001)** | -0.071±0.049 ($p$=0.140) |
| RSA_layer2 | -0.010±0.027 ($p$=0.714) | -0.031±0.033 ($p$=0.355) | **-0.108±0.035 ($p$=0.003)** |
| RSA_layer3 | **0.218±0.034 ($p$<0.001)** | -0.021±0.036 ($p$=0.558) | **0.214±0.062 ($p$=0.001)** |
| geometric_layer1 | **0.040±0.010 ($p$<0.001)** | **-0.487±0.090 ($p$<0.001)** | -0.130±0.149 ($p$=0.368) |
| geometric_layer2 | **0.020±0.007 ($p$=0.004)** | -0.012±0.041 ($p$=0.769) | -0.310±0.230 ($p$=0.173) |
| geometric_layer3 | -0.021±0.012 ($p$=0.075) | **0.191±0.084 ($p$=0.025)** | -0.174±0.112 ($p$=0.120) |
| FBPS_L2_layer1 | 0.009±0.008 ($p$=0.262) | 0.001±0.001 ($p$=0.439) | **0.047±0.008 ($p$<0.001)** |
| FBPS_L2_layer2 | 0.010±0.007 ($p$=0.157) | -0.001±0.001 ($p$=0.156) | -0.009±0.008 ($p$=0.287) |
| FBPS_L2_layer3 | 0.004±0.002 ($p$=0.070) | **-0.099±0.041 ($p$=0.025)** | **0.732±0.215 ($p$=0.002)** |
| FBPS_KL_output | **-0.364±0.014 ($p$<0.001)** | **0.375±0.019 ($p$<0.001)** | 0.055±0.036 ($p$=0.123) |

**Feature Weighting Manipulation Across Different Boost Factor.** When decreasing the foreground features, as the boost factor gets smaller, BACKGROUND-INVARIANCE performances decrease more while OBJECT-DISAMBIGUATION performances increase more in both datasets. Conversely, when decreasing the background features, as the boost factor gets smaller, BACKGROUND-INVARIANCE performances increase more while OBJECT-DISAMBIGUATION performances decrease more in both datasets. However, when increasing the feature importance in either foreground or background subspaces, the performances never deviate significantly from the control experiments. We suspect that this is because when boosting the subspaces, the resulting weights of the linear classifiers are smaller, and thus the regularization that penalizes larger weights will have a weaker effect.

## C.4 Proposed Augmentation Method

**Augmentation using pseudo-masks generated by Segment-Anything (SAM).** We also experiment with using the pseudo-masks generated by SAM to alleviate the need for additional annotations. However, since SAM over-generates masks, we cross-check the pseudo-masks with ground-truth masks by selecting the highest IoU. This process should be easily automated with methods like Lai

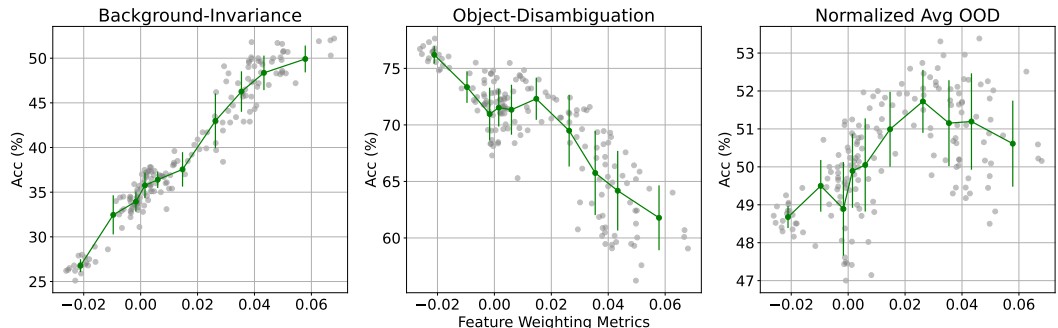

Figure 10: Feature Weighting vs OOD Accuracies in SCENEOBJECT. The three plots show the relationship between feature weighting metrics and BACKGROUND-INVARIANCE, OBJECT-DISAMBIGUATION, and the average OOD accuracies.

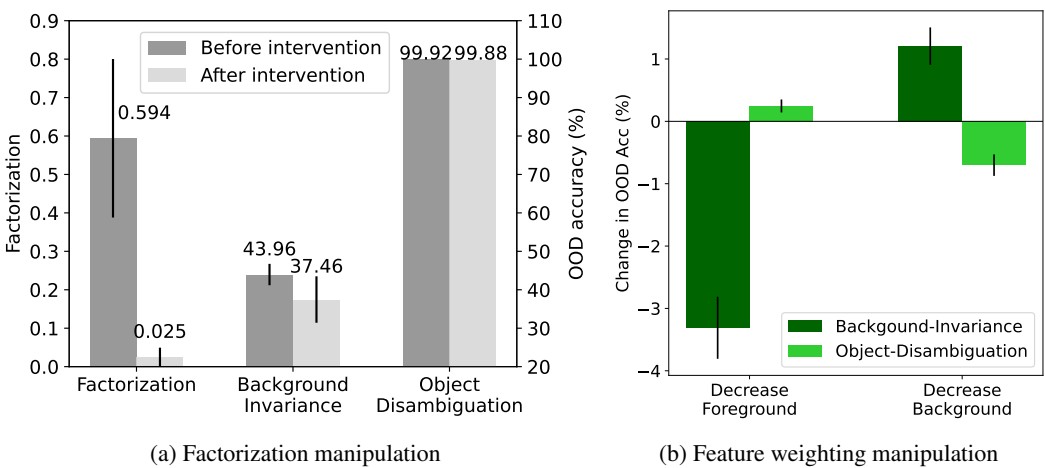

(a) Factorization manipulation           (b) Feature weighting manipulation

Figure 11: Causal intervention results using 10 ERM models in COLOROBJECT.

et al. [22], providing ground for future work to explore. As shown in Table 6, with the generated pseudo-masks, we achieve similar performances as the original method using the ground-truth masks.

**Training size ablation.** We evaluate the effectiveness of our augmentation methods on various training sizes (4k, 8k, 16k) in SCENEOBJECT. The effect size of our data augmentation method is more significant when the data size is smaller. The improvements go up to 4.5% on BACKGROUND-INVARIANCE and 1.1% on OBJECT-DISAMBIGUATION compared to the baseline ERM with 4k training data, while the best baseline (Fish) does not achieve better OOD performances than the baseline.

**Effect of Mixing Weight on OOD Performances.** We show the effect of mixing weight on OOD performances in this section. The two loss items in our proposed augmentation method are combined using a mixing weight (from 0 to 1) so that we can keep the learning rate fixed. We show the impact of mixing weight on BACKGROUND-INVARIANCE, OBJECT-DISAMBIGUATION, and the average accuracies in both datasets in Fig. 13. In both COLOROBJECT and SCENEOBJECT, as the mixing weight increased, BACKGROUND-INVARIANCE accuracy increases while OBJECT-DISAMBIGUATION accuracy decreases gradually. This suggests that the models become more invariant to the backgrounds and rely more on the foreground features. However, the average accuracy exhibits a different pattern. Initially, as the mixing weight increased, the average accuracy showed an upward trend. The accuracy reached a peak and then started to decrease gradually as the mixing weight continued to increase. This finding indicates that there exists an optimal mixing weight that maximizes the overall accuracy of the model on different OOD settings, suggesting the need to find an appropriate weight between foreground and background features.

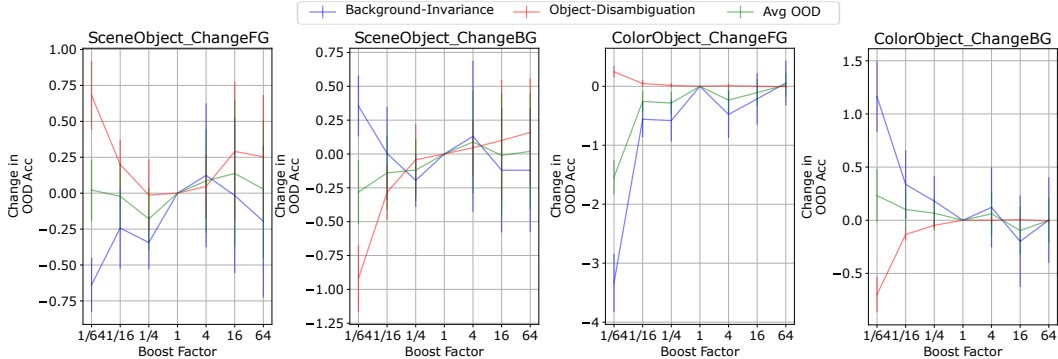

Figure 12: Effect of Boost Factor on OOD Performances.

Table 6: Accuracy of our proposed method with pseudo-masks generated by Segment-Anything (SAM) or with various training sizes (4k, 8k, 16k) on SCENEOBJECT [19]. Our method with pseudo-masks achieves similar generalization performances as the one with ground-truth masks. The effect size of our data augmentation method is more significant when the data size is smaller.

| Method | SCENEOBJECT | | | |
| | ID | OOD1 | OOD2 | Avg OOD |
|---|---|---|---|---|
| ERM | 72.28±0.93 | 35.37±1.97 | 71.93±1.27 | 53.65±0.70 |
| Best Baseline | 73.47±0.79 | **36.91±1.49** | 71.30±0.74 | 54.11±0.61 |
| Ours (pseudo-masks) | **75.00±1.08** | 37.96±2.04 | **73.34±2.09** | **55.65±2.07** |
| Ours (gt-masks) | 74.13±0.48 | 36.83±1.34 | 73.86±1.11 | 55.34±0.51 |
| ERM (8k data) | 66.97±1.05 | 31.29±1.37 | 68.57±0.60 | 49.93±0.99 |
| Best Baseline (8k data) | 67.13±0.74 | 32.19±0.64 | 67.08±0.46 | 49.64±0.55 |
| Ours (8k data) | **71.43±0.89** | **34.52±0.97** | **72.16±1.24** | **53.34±1.11** |
| ERM (4k data) | 60.74±0.89 | 26.27±1.08 | 62.79±1.04 | 44.53±1.06 |
| Best Baseline (4k data) | 61.35±1.60 | 26.77±1.45 | 63.34±0.90 | 45.06±1.18 |
| Ours (4k data) | **63.94±0.51** | **30.72±0.60** | **63.86±0.54** | **47.29±0.57** |

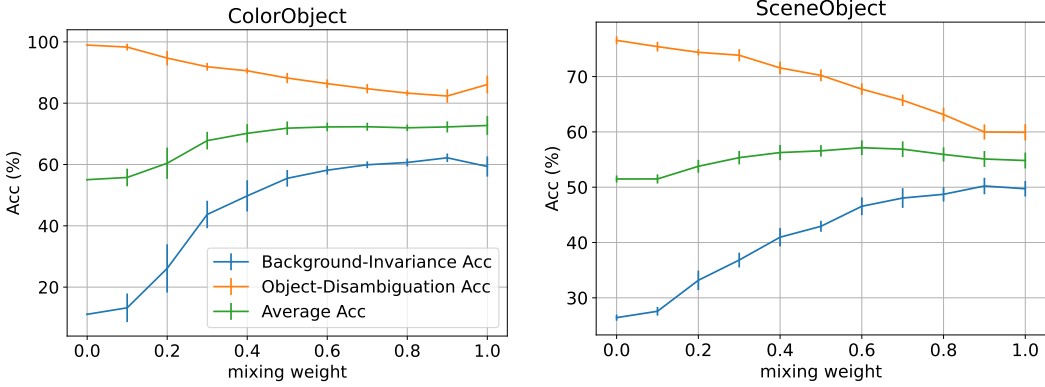

Figure 13: Effect of Mixing Weight on OOD Performances. As the mixing weight increases, the BACKGROUND-INVARIANCE performance increases while the OBJECT-DISAMBIGUATION performance decreases. The average OOD performances either flattens out (as in COLOROBJECT) or peaks and then starts decreasing (as in SCENEOBJECT)

