# OpenReview forum: "Adaptive Contextual Perception: How To Generalize To New Backgrounds and Ambiguous Objects"
_NeurIPS.cc/2023/Conference — NeurIPS 2023 poster_

### Official Review · Reviewer_Qyp3 · 2023-07-04

**Soundness:** 3 good
**Presentation:** 3 good
**Contribution:** 3 good
**Rating:** 5
**Confidence:** 5

**Summary:**

This paper presents an empirical study on out-of-distribution (OOD) generalization in two different settings: contexts that are either beneficial or irrelevant. The authors highlight an interesting finding that models performing well in one setting tend to struggle in the other. They analyze a population of models and demonstrate that those with more factorized representations and appropriate feature weighting achieve improved performance in handling both OOD settings. Additionally, the paper introduces novel data augmentation methods aimed at enhancing model generalization.

**Strengths:**

One of the major strengths of this paper is its detailed analysis of generalization and the identification of factors that influence generalization performance.

**Weaknesses:**

While the paper proposes a data augmentation method, the motivation behind its introduction lacks strong support.

**Questions:**

In section 7, the authors describe the proposed data augmentation method without adequately establishing its connection to the motivation of “adaptively using context to recognize objects in new settings” due to the following reasons: (1) The weight of the training loss is set as fixed hyperparameters and lacks the adaptive nature required to effectively utilize context. (2) The data augmentation method does not provide clear instructions on how context can be effectively employed for object recognition.

**Limitations:**

The method requires additional annotation of the background classes and the segmentation of the foreground objects.

---

> ### Author Rebuttal · Authors · 2023-08-09
>
> > “the authors describe the proposed data augmentation method without adequately establishing its connection to the motivation of “adaptively using context to recognize objects in new settings””
>
> We establish the connection between our analysis and our proposed method in the introduction “In order to encourage model factorization, we augment the data with random-background and background-only images, and we weight the corresponding objective terms to encourage appropriate feature weighting of the foreground and background information” (Lines89-91). Our proposed method does not directly manipulate the representation geometry, but we reason that our proposed method encourages the models to represent the foreground and background in the right way (Lines367-369).
>
> > “The weight of the training loss is set as fixed hyperparameters and lacks the adaptive nature required to effectively utilize context.”
>
> We want to clarify what “adaptive” means here. Adaptive means the model uses context when the object features are ambiguous but ignores the context when the object is easily recognizable (Lines35-38), which is viable with a fixed weight. And this is evidenced by the improved performances in two OOD settings, which is what our method achieves. It is not the case that making the weight variable would make the method "adaptive". The weight only shows how the evidence is combined. And we argue there is an optimal weight depending on the domains for optimal generalization (see Line 338-340 & Figure 4b).
>
> > “The method requires additional annotation of the background classes and the segmentation of the foreground objects.”
>
> We appreciate the concern, and we agree that it would be ideal for the method to not require any additional human annotations. Please refer to the General Response and pdf Table R1, where we add results using SegmentAnything pseudo-masks rather than our ground-truth masks on ColorObject and SceneObject. We are pleased to see performance improvements with SegmentAnything masks that are similar to our original results (though note we require heuristic cross-checking with the ground-truth masks, which we believe could be alleviated with more controllable segmentation methods like https://arxiv.org/abs/2308.00692). This result should provide a pathway for future work to obtain good foreground/background separation masks without the need for additional human annotation, enabling the application of our data augmentation method to improve generalization to background-invariance and object-disambiguation tasks.

---

> > ### Comment · Area_Chair_aAg8 · 2023-08-18
> > **Engage in the discussion with the authors**
> >
> > Dear Reviewer,
> >
> > The author has provided responses to your questions and concerns. Could you please read their responses and ask any follow-up questions, if any?
> >
> > Thank you!

---

### Official Review · Reviewer_FMxu · 2023-07-04

**Soundness:** 3 good
**Presentation:** 4 excellent
**Contribution:** 3 good
**Rating:** 7
**Confidence:** 4

**Summary:**

This paper examines the influence of context on visual recognition capabilities. The authors distinguish two regimes, one where background context can help disambiguate objects and another one where the object information is orthogonal to the background information. The study shows that models can thrive in one regime but fail to extrapolate to the other regime. The paper then suggests that factorizing object and background information can help in out-of-distribution generalization and proposed data augmentation methods to help models generalize.



**Strengths:**

The work is clearly written and presents a nice systematic examination of different aspects of how context can impact visual recognition.

The work nicely puts the emphasis on generalization to OOD, which is not always tested in other context-aware models.

Figure 3 is nice and clearly demonstrates the trade-off between invariance and the ability to use contextual features.


**Weaknesses:**

No major weaknesses noted, but see questions below.

**Questions:**

The authors distinguish situations where context can help recognition or where context is irrelevant and the desired output is invariance to contextual noise. Context can also hurt performance (see e.g., Bomatter et al ICCV 2021 and multiple studies on incongruent context conditions, summarized in Oliva and Torralba, TICS 2007).

The addition of feature weights and especially the transition to All features lead to small improvements in Table 1. All the other entries are clearly justified and critical for the model. Similar comments apply to Fig. 4b. The enhancement is small.

How is the pareto frontier computed in Fig. 3 and why is it called a frontier if there are points that can trespass this line?


**Limitations:**

There is minimal discussion of limitations in the current version.

---

> ### Author Rebuttal · Authors · 2023-08-09
>
> > “Context can also hurt performance.”
>
> We thank the reviewer for pointing out the relevant works. We believe that the referred works are consistent with ours and will add them to our references. It is true that context can also hurt performance for both models (as observed in our experiments too) and humans. Although the ideal behavior in the Background-Invariance setting is to ignore the background completely, we argue that there needs to be some assumption about what the test distributions will look like, in order to decide how sensitive the model should be to the backgrounds. This is because models and humans need to deal with both Background-Invariance and Object-Disambiguation settings. Our Pareto plots show that models should not be completely invariant to the background, because this would drastically lower performance in an object-disambiguation setting.
>
> > “The addition of feature weights and especially the transition to All features lead to small improvements in Table 1… Similar comments apply to Fig. 4b. The enhancement is small.”
>
> We want to clarify that in Table 1, the row for Feature Weighting metrics do not include Factorization metrics. That is, the Factor. row does not include Feature Weighting in the regression, nor does the Feature Weighting row include the Factorization metric in the regression. In other words, both feature factorization and weighting on their own are both strong predictors. We apologize for any confusion in the table formatting and will clarify this further. This said, the transition to the All row (final row) does show that these two variables are fairly collinear, with each providing a relatively smaller amount of information that the other does not contain. However, the first takeaway from this table is the very large improvement over the ID baseline, resulting in high R^2 values above 0.85. Past work has not been able to obtain _any improvement_ over ID, while we're getting quite large improvements in R2.
>
> In Figure 4, we wanted to show causal evidence for our hypotheses about OOD generalization, which has not been done before, to our knowledge. We believe the changes in accuracy of up to 3 points are meaningful, especially when considered alongside the high R^2 values of our regressions. The results are also all statistically significant (Lines333-337), so we believe this experiment supports the overall argument for the causal importance of these features.
>
> > “How is the pareto frontier computed in Fig. 3 and why is it called a frontier if there are points that can trespass this line?”
>
> We fit a polynomial function to the outer edge of all the data points as the Pareto frontier. Since the data is noisy, we would see some points trespassing this line. This is not the ground truth frontier, which no points should pass through. Considering the large size of the model population, we believe the ground truth frontier would be in a similar shape as our approximate frontier, showing a similar kind of trade-off. We can add these details to the final paper.

---

> > ### Comment · Area_Chair_aAg8 · 2023-08-18
> > **Engage in the discussion with the authors**
> >
> > Dear Reviewer,
> >
> > The author has provided responses to your questions and concerns. Could you please read their responses and ask any follow-up questions, if any?
> >
> > Thank you!

---

### Official Review · Reviewer_WSbu · 2023-07-06

**Soundness:** 4 excellent
**Presentation:** 4 excellent
**Contribution:** 3 good
**Rating:** 5
**Confidence:** 4

**Summary:**

This work investigates how visual models leverage background and foreground information for out-of-distribution (OOD) generalization. The authors trained a large number of models and evaluate their performance in two OOD settings. They find that there is a tradeoff for the models in these two OOD settings as they need to balance how much foreground and background are separately used during the recognition task. Multiple analyses are further conducted to better understand this tradeoff. The authors find that “factorized representations and appropriate feature weighting are more successful in handling” the OOD tests. They also present experiments supporting the causal influence of these factors. Finally, they propose a new method to train the models through adding more background and foreground related augmentations. This method yields a better model compared to existing methods.

**Strengths:**

The paper is well written. The logic is clear. The results are clearly presented. I enjoy reading the paper.

The main idea in this paper is analyzing the models’ performance in two separate but relevant OOD tests: OBJECT-DISAMBIGUATION and BACKGROUND-INVARIANCE, where the background information is either beneficial or irrelevant. This idea is also innovative. To thoroughly investigate the performance of different algorithms, the authors train a lot of models, which makes the results general.

The regression and probing analysis is also neat. After leveraging the results from this analysis, the authors also create a new method that yields a better model.


**Weaknesses:**

My biggest worry is that the results reported in this work is just reflecting the models trained on small amount of data and cannot generalize to models trained with much more data. The training set of the models only contain 16000 images, not mentioning that many of these images are generated from the same object. So these images contain very limited diversity across objects and natural background. State-of-the-art models in recognition are trained with at least 2 magnitude more images, not mentioning very recent models trained on almost billion scale images (like Segment-Anything). Is the tradeoff observed in this work just a result due to not enough training data? When there are more data, maybe the model can achieve the optimal performance in both benchmarks. Then how would the finding in this paper be general to more realistic settings that are used in actual applications?

In addition to this worry, the proposed new method is also over-engineered for the constructed benchmarks. The new method is simply augmenting the data with background and foreground operations, with clear implications to benefit the constructed benchmarks. How would this augmentation benefit other OOD settings? Is this augmentation useful for training on more images?


**Questions:**

See comments in the weakness.

**Limitations:**

yes

---

> ### Author Rebuttal · Authors · 2023-08-09
>
> > “Is the tradeoff observed in this work just a result due to not enough training data? When there are more data, maybe the model can achieve the optimal performance in both benchmarks.”
>
> The two standard settings (ColorObject and SceneObject) we test on for fair comparison with prior works are indeed small. While we aren’t able to conduct experiments with data of the SAM size (1B images) due to academic resource constraints, we are able to produce “scaling laws” results in the form of training size ablations. Shown in the General Response and pdf Fig. R1, we find a clear trade-off between the two OOD settings regardless of how much training data is available to the model. Note the frontier moves “outward” as the training size increases, since models become more accurate on both tests. We expect the same trade-off to hold with more training data with the frontiers moving further outwards.
>
>
> > “the proposed new method is also over-engineered for the constructed benchmarks… How would this augmentation benefit other OOD settings?”
>
> We test our method on two standard datasets with metrics in three domains, in-distribution, Background-Invariance, and Object-Disambiguation. Past published work has proposed new methods with only these datasets, and using only one of our two OOD tests (including the strongest baseline, Fish [39]). We believe this is a good proof of concept for the new data augmentation method. It would be interesting to test other domains shifts, but we consider that out of scope for our paper.
>
> > “Is this augmentation useful for training on more images?”
>
> While we are not able to conduct large scale experiments on larger sizes than 16000 training points, we provide an ablation of our data augmentation method across different training sizes in the General Response and pdf Table R1. We do observe that the boost to OOD performance from the data augmentation method is larger when there is less training data available, although for now we cannot say for certain that the effect would entirely dissipate with larger training sizes, especially given the fundamental mismatch between the in-distribution training data and the Background-Invariance and Object-Disambiguation tests. Moreover, we believe one additional benefit of the data augmentation method is that it allows one to bias the model toward higher performance on either the Background-Invariance _or_ Object-Disambiguation tests, by adjusting the weight on the objective terms in the loss. This is a new benefit of the method that would enable one to tailor their model to perform better in one setting or the other, based on an expectation of how the test-time distribution will look for a model in deployment.

---

> > ### Author Response · Authors · 2023-08-17
> >
> > Thank you for your comment! PDFs for all papers were temporarily unavailable, presumably due to some OpenReview system issues. Our PDF is now accessible again and can be found at the bottom of our general response (https://openreview.net/forum?id=7JuReDmGSL&noteId=pLEPyZJhLW). Please let us know if you have any further questions.

---

> > > ### Comment · Reviewer_WSbu · 2023-08-18
> > >
> > > Thanks for the response and the new results. First, from Fig. R1a, I agree with the authors that the scale does not critically influence the results reported in this work (“tradeoff exists”). But the Swin results are surprising to me. In the Fig. R1b, Swin got much better results than the ResNets, especially on the Object-Disambiguation task. Theoretically, the tradeoff does exist. But empirically, the right-most point on that figure seems to work just fine to me, as its y-value is only ~0.05 lower than the highest point, but its x-value is much better than the other points. Is the tradeoff still worth being explored in this case? Especially, this is only on the scale of 16000 images. If the models are trained with more images (like just 10 times more), will the right-most point achieve just good enough performance on both benchmarks, which makes the discussion of this tradeoff unnecessary?

---

> > > > ### Author Response · Authors · 2023-08-19
> > > >
> > > > > In Fig. R1b…empirically, the right-most point on that figure seems to work just fine to me, as its y-value is only ~0.05 lower than the highest point, but its x-value is much better than the other points. Is the tradeoff still worth being explored in this case?
> > > >
> > > > It’s true that Swin does quite well on our task. The point you mention gets about 74% on the background-invariance task and 85% on the object-disambiguation. What we’d like to emphasize is our argument in lns236-238, which is that whether this is a good point on the Pareto curve depends on the environment expected at deployment. If you deploy a model in a setting with no ambiguous objects, then this lower-right point would be the best one to pick. But, for example, a self-driving car vision model may face an environment where objects are typically in their “usual” background, but objects are often occluded, darkened, small, and far away, etc. If this were the environment, then it would be better to pick a point higher up on the curve, that does better at object disambiguation (at some trade-off with background-invariance). And we believe the trade-off will be worth exploring when performance swings by as much as 5% – a difference in error rates of 5% in a model deployed at scale is huge.
> > > >
> > > > > …If the models are trained with more images (like just 10 times more), will the right-most point achieve just good enough performance on both benchmarks, which makes the discussion of this tradeoff unnecessary?
> > > >
> > > > This is a good question. Will models reach 100% OOD generalization, and if they do, will there no longer be any trade-off in object-disambiguation and background-invariance? We suppose, though we do not know, that it is possible. Perhaps the Pareto frontier will expand with better and better models until there is a point at 100% accuracy for both OOD tests. In this case, no trade-off will have to be made! However, the model performances are not close to 100% even with strong transformer models. We think it is very likely that, in complex vision environments like ImageNet, models will continue to have sub-100% OOD generalization even with significantly more data, in which case the trade-off remains a pressing problem.

---

### Official Review · Reviewer_D3vS · 2023-07-07

**Soundness:** 3 good
**Presentation:** 3 good
**Contribution:** 3 good
**Rating:** 5
**Confidence:** 4

**Summary:**

This work investigates how vision models use background information in various contexts and find that models that are more invariant to backgrounds are less able to use the background to disambiguate. A new objective function and augmentations are proposed to control the balance between ignoring the background and using it for disambiguation. It is hypothesized that in order to make adaptive use of background as in biological vision, computer vision models must have factorized (orthogonal) object vs background representations.

**Strengths:**

1. Clear presentation and motivation.
2. While many works have investigated the role of background vs foreground, much of the literature has focused on over-reliance on the background leading to spurious correlations, though background can also disambiguate. While this is still "known to the field", this  systematic study can still be helpful to the community.

**Weaknesses:**

1. There is some re-treading of scientific points from Xiao et al, 2020.

2. The architecture is limited to Wide ResNet-28 and MLP for synthetic datasets. Transformers are known to make adaptive use of context (and are an important, broadly applicable architecture, arguably more so than ResNets).  How do these findings change when using vision transformers is an important unanswered question and a limitation. Also unknown is how the findings change with respect to model scale.

3. The augmentations proposed are possible on toy or synthetic datasets, but the need for dense annotations like segmentation masks for foreground objects makes the approach not really practical - especially for more complex scenes, with many objects and scales. If the authors really want to argue that the augmentation they propose is a new *methodological contribution*, then they will need to demonstrate it's feasibility on more practical datasets with more complex scenes. If the authors could please elaborate on this in the response, that would be great.

**Questions:**

See weaknesses.

**Limitations:**

Yes.

---

> ### Author Rebuttal · Authors · 2023-08-09
>
> > “There is some re-treading of scientific points from Xiao et al, 2020.”
>
> Xiao et al. 2020 focus on showing that models over-rely on background. In fact, they argue that models rely on background too much, writing that models “exploit background correlations” and hoping for models to exhibit “robustness to misleading background signals.” We argue instead that models need both foreground and background by proposing a novel mechanism (feature factorization + appropriate weighting), which is supported by our regression and causal analysis and augmentation method.
>
>
> > “How do these findings change when using vision transformers is an important unanswered question and a limitation.”
>
> Thanks for the suggestion. We have extended our results to include Vision Transformers (Swin), focusing first on the tradeoff between generalization to background-invariance and object-disambiguation tests. As shown in the General Response and pdf Fig. R1, we are pleased to find that Transformers show the same tradeoff in generalization as ResNets. We will add these additional results in the final version.
>
>
> > “need to demonstrate it's feasibility on more practical datasets with more complex scenes.”
>
> We appreciate the suggestion, and we agree that it would be ideal to showcase the data augmentation method on more practical datasets. Please refer to the General Response and pdf Table R1, where we add results using Segment Anything masks rather than our ground-truth masks on ColorObject and SceneObject. We are pleased to see performance improvements with SegmentAnything masks that are similar to our original results (though note we require heuristic cross-checking with the ground-truth masks, which we believe could be alleviated with more controllable segmentation methods like https://arxiv.org/abs/2308.00692). Since the Segment Anything Model obtains impressive segmentation results with complex naturalistic images, we are hopeful that future work can build on our method to improve how SOTA models generalize across background-invariance and object-disambiguation tests with naturalistic data.

---

> > ### Comment · Area_Chair_aAg8 · 2023-08-18
> > **Engage in the discussion with the authors**
> >
> > Dear Reviewer,
> >
> > The author has provided responses to your questions and concerns. Could you please read their responses and ask any follow-up questions, if any?
> >
> > Thank you!

---

### Author Rebuttal · Authors · 2023-08-09

We thank the reviewers for their time and effort in reading our paper. We are glad to see that reviewers view the paper to be “helpful to the community” (D3vS) and “innovative” (WSbu), with a “systematic examination” (FMxu) and “detailed analysis…of factors that influence generalization performance” (Qyp3). In our individual responses to each review, we try to address any noted questions about the paper.

Here, we wish to highlight a few extensions of our experiments with additional training sizes, model architectures, and SAM-generated image masks, all of which are consistent with our original results and demonstrate robustness across additional experiment setups.

**Training Size Ablations and Transformer Results.** First, to strengthen our analysis results, we provide OOD Pareto frontier plots for (a) models with varying amounts of training data and (b) Vision Transformers (Swin) in addition to ResNets. See Fig. R1 in the response pdf for these results, respectively. Our goal here is to demonstrate that models must tradeoff between background-invariance and object-disambiguation performance in general, regardless of training data scale or model architecture. Our results are consistent with our original experiments using ResNet and 16k training points: all models show a clear tradeoff between generalization to background-invariance and object-disambiguation tests.

**Segment Anything Model Experiments.** Second, to provide a stronger proof of concept for our data augmentation method, we present a new heuristic masking method based on the Segment-Anything model. We show that this model can automatically segment images into foreground and background, which enables our data augmentation method to apply generically without the need for ground-truth foreground/background masks. Using pseudo-masks generated by Segment-Anything, we achieve comparable results to using the ground-truth masks (see Table R1). In addition, new segmentation datasets like SA-1B provide billions of segmentation masks for tens of millions of naturalistic images. Thus we believe our method can provide fertile ground for future work to explore with Segment-Anything and more complex scenes.

---

### Decision · Program_Chairs · 2023-09-21

**Decision:**

Accept (poster)

**Comment:**

All reviewers reached a consensus to accept the paper. The authors are encouraged to incorporate all the feedback and suggestions from the reviewers into the final version of the manuscript.